# Characterization of IgA Deposition in the Kidney of Patients with IgA Nephropathy and Minimal Change Disease

**DOI:** 10.3390/jcm9082619

**Published:** 2020-08-12

**Authors:** Won-Hee Cho, Seon-Hwa Park, Seul-Ki Choi, Su Woong Jung, Kyung Hwan Jeong, Yang-Gyun Kim, Ju-Young Moon, Sung-Jig Lim, Ji-Youn Sung, Jong Hyun Jhee, Ho Jun Chin, Bum Soon Choi, Sang-Ho Lee

**Affiliations:** 1Department of Medicine, Graduate School, Kyung Hee University, Seoul 02447, Korea; minime12@naver.com; 2Division of Nephrology, Department of Internal Medicine, Kyung Hee University Hospital at Gangdong, Seoul 05278, Korea; 01love14@hanmail.net (S.-H.P.); chltmfrl92@khnmc.or.kr (S.-K.C.); ha-ppy@daum.net (S.W.J.); apple8840@hanmail.net (Y.-G.K.); kidmjy@hanmail.net (J.-Y.M.); 3Division of Nephrology, Department of Internal Medicine, Kyung Hee University Medical Center, Seoul 02447, Korea; aprilhwan@naver.com; 4Department of Pathology, Kyung Hee University Hospital at Gangdong, Kyung Hee University, Seoul 05278, Korea; sungjig@khu.ac.kr; 5Department of Pathology, Kyung Hee University Hospital, Kyung Hee University College of Medicine, Seoul 02447, Korea; jyune15@gmail.com; 6Division of Nephrology, Department of Internal Medicine, Gangnam Severance Hospital, Yonsei University College of Medicine, Seoul 06273, Korea; JJHLOVE77@yuhs.ac; 7Department of Internal Medicine, Seoul National University Bundang Hospital, Seongnam 13620, Korea; mednep@hanmail.net; 8Department of Internal Medicine, College of Medicine, Seoul National University, Seoul 03080, Korea; 9Division of Nephrology, Department of Internal Medicine, College of Medicine, The Catholic University of Korea, Seoul 06591, Korea; sooncb@catholic.ac.kr

**Keywords:** IgA nephropathy with minimal change disease, nephrotic syndrome, galactose-deficient IgA1, KM55, double immunofluorescent staining

## Abstract

Approximately 5% of patients with IgA nephropathy (IgAN) exhibit mild mesangial lesions with acute onset nephrotic syndrome and diffuse foot process effacement representative of minimal change disease (MCD). It is not clear whether these unusual cases of IgAN with MCD (IgAN-MCD) are variant types of IgAN or coincidental deposition of IgA in patients with MCD. In a retrospective multicenter cohort study of 18 hospitals in Korea, we analyzed 46 patients with IgAN-MCD. Patients with endocapillary proliferation, segmental sclerosis, and crescent were excluded, and the clinical features and prognosis of IgAN-MCD were compared with those of pure MCD. In addition, we performed galactose-deficient IgA1 (KM55) staining to characterize IgAN-MCD. Among the 21,697 patients with glomerulonephritis enrolled in the database, 46 patients (0.21%) were diagnosed with IgAN-MCD, and 1610 patients (7.4%) with pure MCD. The 46 patients with IgAN-MCD accounted for 0.6% of primary IgAN patients (*n* = 7584). There was no difference in prognosis between patients with IgAN-MCD and those with only MCD. IgA and KM55 showed double positivity in all patients with IgAN-MCD (*n* = 4) or primary IgAN (*n* = 5) under double immunofluorescent staining. However, in four patients with lupus nephritis, mesangial IgA was deposited, but galactose-deficient-IgA1 (Gd-IgA1) was not. These findings suggest that IgAN-MCD is a dual glomerulopathy in which MCD was superimposed on possibly indolent IgAN. We confirmed by KM55 staining that IgAN-MCD is true IgAN, enabling better characterizations of the disease. Furthermore, IgAN-MCD shows a good prognosis when treated according to the usual MCD treatment modality.

## 1. Introduction

IgA nephropathy (IgAN) is the most prevalent glomerulonephritis (GN) in Asia, and about 30% of patients progress slowly to end-stage renal disease within 30 to 40 years [1,2]. Patients with IgAN exhibit a variety of clinical manifestations, including synpharyngitic hematuria, subnephrotic range proteinuria, and gradually decreasing renal function. The kidney biopsy specimens of some patients show rapidly progressive renal deterioration with crescent formation. Nephrotic syndrome (NS) has been reported to occur in about 5% of patients with IgAN [1]. The presence of nephrotic range proteinuria usually indicates advanced IgAN with severe glomerular damage [3]. Rarely, some patients with IgAN have mild mesangial lesions with acute onset NS and diffuse foot process effacement resembling minimal change disease (MCD) [4,5]. These unusual cases of IgAN with MCD (IgAN-MCD) have been previously reported on [6,7]. Some studies have reported these patients to be suffering from a variant type of IgAN, while others have reported this condition as a coincidental deposition of IgA in patients of MCD [4,5,8,9,10,11]. 

According to the multi-hit hypothesis accepted as the pathogenesis of IgAN, overproduction of galactose-deficient IgA1 (Gd-IgA1) and subsequent recognition of Gd-IgA1 by IgG or IgA autoantibodies results in the formation of immune complexes, and their subsequent mesangial deposition results in the activation of secondary immune mediators and glomerular injury. Gd-IgA1 is considered to be the core component at the beginning of IgAN pathogenesis [12]. In the past, the lectin-dependent snail helix aspersa agglutinin (HAA) assay, which measures serum Gd-IgA1 using the ability of Gd-IgA1 to adhere to lectin, presented a crucial problem of bioactivity and stability. Commercially available KM55 is a monoclonal antibody that interacts with the hinge region in human Gd-IgA1, and is known to be an easy and reliable tool for measuring Gd-IgA1 levels [13]. 

There has been much controversy over whether these unusual cases of IgAN with MCD (IgAN-MCD) are variant types of IgAN or a coincidental deposition of IgA in patients with MCD. Moreover, the pathogenesis of patients classified as IgAN-MCD has yet to be clearly investigated. Therefore, we assessed the clinical features and prognosis of IgAN-MCD in this retrospective cohort. Further, kidney samples from IgAN-MCD patients were stained with KM55 to determine whether IgA deposited in the mesangium shares the same pathophysiology as that deposited in primary IgAN.

## 2. Materials and Methods

### 2.1. Patients

In a KoGNET (Korean Glomerulo Nephritis Study) Group database in Korea, a retrospective multicenter cohort study listed 21,697 glomerulonephritis (GN) patients from 18 different centers from January 1979 to May 2019, among whom we collected and analyzed 46 patients with IgAN-MCD and another 1610 patients with MCD for comparison. 

Diagnostic criteria are as follows: 

1. IgAN-MCD: In addition to the typical diagnostic criteria of IgAN (diffuse IgA-dominant depositions in the glomerular mesangial region and mesangial electron-dense deposition revealed by electron microscopy), we excluded those diagnosed with secondary IgAN (such as systemic lupus erythematosus, Henoch-Schönlein purpura, psoriasis, rheumatoid arthritis, liver cirrhosis, and celiac disease). IgAN-MCD also presents with minimal glomerular changes (M0 or M1, E0, S0, C0 specifically excluding E1, S1, and C1-2 according to the Oxford classification of IgAN, but including patients with global glomerulosclerosis in some glomeruli, depending on age) under light microscopy and diffuse effacement of podocyte foot processes (>80% of the capillary surface area involved) under electron microscopy. 2. MCD: Patients with an absence of prominent glomerular lesions as observed by light microscopy, trace C3, IgM, or negative deposition of IF staining, diffuse effacement of podocyte foot processes, and absence of electron-dense deposits as revealed by electron microscopy manifested as NS were considered. Patients with secondary nephritis and other chronic diseases (such as diabetic nephropathy, systemic amyloidosis, and hepatitis B-associated nephropathy) were excluded. 

### 2.2. Preparation of Propensity Score-Matched Pairs 

To analyze the effects of IgAN-MCD and MCD on clinical outcomes, nearest neighbor matching, a propensity score matching method, was performed. Age, sex, body mass index (BMI), diabetes mellitus (DM), hypertension (HTN), and urine protein to creatinine ratio (UPCR) were determined as covariates (Table 1). The standardized difference was measured to determine the balance of covariates. Ultimately, 46 pairs of IgAN-MCD and 138 pairs of MCD patients were analyzed after matching. All five variables were matched using a standardized difference of less than 0.1. 

### 2.3. Evaluation of Treatment Efficacy

We compared the clinical outcome of IgAN-MCD and MCD patients at 6 months, 12 months, and at final follow-up after biopsy: CR was defined as UPCR < 0.5 g/g and serum albumin > 3.5 g/dL; NR was defined as a UPCR > 3.5 g/g, a decline < 50% or increase of baseline value, and/or an Scr elevation of > 50% from baseline.

### 2.4. Double Immunofluorescence Staining of Kidney Biopsy Tissues

Kidney biopsy specimens were obtained from three IgAN-MCD patients from Kyung Hee University Hospital at Gangdong, and one IgAN-MCD patient from Kyung Hee University Medical Center registered to the KoGNET database. We obtained informed consent from all four patients and the approval of the Research Ethics Review Committees of Kyung Hee University Hospital at Gangdong and Kyung Hee University Medical Center. In addition, five specimens from IgAN patients and four specimens from lupus nephritis cases from Kyung Hee University Hospital at Gangdong were obtained as positive and negative controls, respectively. Immunofluorescent staining of Gd-IgA1 in glomerular tissues was performed. Paraffin-embedded sections of 3-μm thickness were prepared for staining. After deparaffinization by a series of xylene/ethanol and rehydration, antigen retrieval with 0.05% bacterial protease subtilisin A (Sigma-Aldrich, Tokyo, Japan) was performed at 37 °C for 30 min. Then, the samples were rinsed with distilled water and blocked with 1% bovine serum albumin (Cellnest, cat no. CNB102-0100)/phosphate-buffered saline (PBS) at room temperature for 30 min, followed by incubation with KM55 (200 μg/mL) at 4 °C for 18 h. After several washes with PBS and triethanolamine-buffered saline, containing 0.05% tween-20 (TBST), the samples were incubated with Alexa Fluor 555-conjugated goat anti-rat IgG antibody (1:1000 diluted; Life Technologies, Carlsbad, CA, USA) at 37 °C for 60 min. Samples were then washed with PBS/TBST and incubated with FITC-conjugated polyclonal rabbit anti-human IgA antibody (100 μg/mL; Dako Japan) at 37 °C for 60 min. After washing with PBS/TBST, the slides were sealed in Fluoromount (Diagnostic BioSystems, CA, USA). Finally, fluorescence was observed using confocal microscopy (LSM-700; Carl Zeiss Microscopy GmbH, Jena, Germany), and images were analyzed in Zen 2011 software. 

### 2.5. Statistical Analysis

Normally distributed variables were expressed as the mean ± SD, and differences among groups were analyzed by *t*-test. Non-parametric variables were expressed as median (range or interquartile range) and compared using the Mann–Whitney test. Categorical variables were expressed in percentages and compared using Pearson χ^2^-test or Fisher’s exact test. Cumulative probabilities of survival and renal endpoints were calculated using Kaplan–Meier methods and compared using the log-rank test. All *p*-values < 0.05 were considered statistically significant. Statistical analyses were performed using SPSS 22.0 statistical software and GraphPad Prism 8.0.

## 3. Results

### 3.1. Incidence of IgAN with MCD

Patients were selected from the 21,697 GN patients registered in the KoGNET database according to the flow diagram shown in Figure 1. A total of 7584 cases (35.0%) with IgAN were enrolled, and 1610 cases (7.4%) with MCD were enrolled. A total of 56 patients with IgAN-MCD were initially selected from the cohort. Of these, nine patients were excluded because of E1, S1, and C1-2, according to the Oxford classification. In addition, one patient was excluded because of hepatocellular carcinoma. Forty-six patients (0.21% of all GN) met the diagnostic criteria of IgAN-MCD. There were 2041 patients diagnosed with MCD, and a total of 265 patients with secondary nephritis and other chronic diseases (diabetic nephropathy, systemic amyloidosis, and hepatitis B-associated nephropathy) were excluded. In addition, 166 patients with malignancy were excluded, and 1610 patients of MCD were eventually analyzed (Figure 1). Forty-six patients with IgAN-MCD accounted for 0.6% of primary IgAN patients (*n* = 7584) and 2.9% of primary MCD patients (*n* = 1610).

### 3.2. Clinical Feature of the Patients with IgAN-MCD

Compared to the patients with pure MCD, patients with IgAN-MCD had higher BMI (25.85 ± 3.59 vs. 24.12 ± 4.09 kg/m^2^, *p* = 0.007) and proteinuria (7.45 ± 5.05 vs. 6.04 ± 5.33 g/g, *p* = 0.032), and lower levels of serum albumin (2.22 ± 0.81 vs. 2.74 ± 1.16 g/dL, *p* = 0.003). Thus, age, sex, BMI, DM, HTN, and UPCR were determined as covariates, and 1:3 matching was performed by nearest neighbor matching, a propensity score matching method. After propensity score matching using the variables listed in Table 1, there were no significant differences in patient characteristics among the matched pairs of IgAN-MCD patients and MCD patients (Table 1). 

### 3.3. Prognosis of the Patients with IgAN-MCD

After propensity score matching, 46 IgAN-MCD and 138 MCD patients were compared. The median follow-up durations in IgAN-MCD and MCD patients were 5.6 (interquartile range: 2.2–9.8) and 5.8 (interquartile range: 3.0–8.7) years, respectively. The follow-up period was limited to 15 years. The proportions of CR in biopsy samples obtained at 6 months, 12months, and final follow-up of IgAN-MCD and MCD patients were 72.7%, 83.3%, and 82.1% and 73.3%, 75.0%, and 66.7%, respectively (*p* > 0.05). No patients in the two groups had end-stage renal disease (ESRD) except for one death with MCD. The patient died of metastatic mantle cell lymphoma three years after diagnosis with MCD. The prognoses of the two groups were favorable, but were without significant differences (Figure 2).

### 3.4. Characterization of IgA Deposition in Patients with IgAN-MCD

To characterize IgA deposited in the mesangium of patients with IgAN-MCD, we obtained kidney tissues from four IgAN-MCD, five IgAN (positive control), and four lupus nephritis (negative control) patients for double IF staining of IgA and Gd-IgA1. Gd-IgA1 was localized in a similar pattern as IgA in the glomeruli of all IgAN-MCD patients. Although the intensity of staining differed among the patients, Gd-IgA1 was distributed mainly in the mesangium, and its localization coincided with the sites of IgA deposition (Figure 3). Glomerular Gd-IgA1 and IgA were positive in all IgAN specimens. However, glomerular Gd-IgA1 was negative in all lupus nephritis specimens, regardless of IgA positivity (Figure 4). These findings suggested that the pathophysiology of IgA deposition in IgAN-MCD patients was common with those of IgAN patients in terms of the deposition of under-galactosylated IgA1. However, the clinical course of IgAN-MCD patients resembled that of MCD patients more closely than that of IgAN patients, which indicates that MCD is superimposed over indolent IgAN. Thus, these findings support that IgAN-MCD does not involve coincidental normal IgA entrapment in MCD patients, but superimposed glomerulopathy with MCD in patients with indolent IgAN.

## 4. Discussion

Since the emergence of IgAN-MCD [5,9,14], there has been much controversy as to whether IgAN-MCD is a variant form of IgAN or simply ordinary IgA deposition in MCD patients. In the present study, 46 rare cases of IgAN-MCD were identified, representing an incidence of 0.21% among all GN patients enrolled in the KoGNET database. The clinical manifestation and prognosis of IgAN-MCD did not differ from those of MCD. Kidney tissue samples of IgAN-MCD patients were stained using KM55, a monoclonal Ab against Gd-IgA1. IgA deposited on the glomeruli of IgAN-MCD patients was identified as under-galactosylated IgA1, not normal IgA.

IgAN-MCD is a rare disease, and only a limited number of studies have been conducted on IgAN-MCD patients. This study identified IgAN-MCD patients by analyzing over 20,000 patients of GN on a large scale. There are few reports of prevalence and clinical features of IgAN-MCD. In a study on 1407 patients with IgAN combined with other glomerular diseases, only eight patients (0.6%) were diagnosed with IgAN-MCD [15]. Regarding the prognosis of IgAN-MCD, a previous study included a follow-up of 77 patients of IgAN-MCD patients for more than three years and showed clinical outcomes comparable with those of MCD patients [16]. Kim et al. reported that unless complete or partial remission was achieved, the prognosis of IgAN with NS was unfavorable [17]. However, the study differed from our study, as it excluded patients with only crescentic GN and included patients with more aggressive GN with endocapillary proliferative and sclerosing GN as well as mild mesangial proliferative GN. Therefore, patients with NS due to advanced IgAN were included in the IgAN with NS group, and hence, the prognosis of this group would be worse than that of the group with pure IgAN-MCD.

By confirming positive Gd-IgA1 staining in the IgAN-MCD glomeruli, we demonstrated that IgA deposition is not coincidental, but pathologically important. Hitoshi et al. reported that glomerular Gd-IgA1 was specifically detected in IgAN and IgA vasculitis but not in other renal diseases, including secondary IgAN due to hepatic diseases. Especially in the cases of lupus nephritis accompanied by glomerular IgA deposition, mesangial Gd-IgA1 staining was negative [18]. These findings corroborate the present results. Taken together, we can infer that Gd-IgA1 plays a role in the pathogenesis of IgAN-MCD, as well as IgAN.

KM55 staining for Gd-IgA was negative in almost all the glomeruli of patients with lupus nephritis except for one patient. Some of the glomeruli of one patient with lupus nephritis showed focal staining of Gd-IgA1 with weak intensity. On the other hand, KM55 was stained throughout all glomeruli in patients with IgAN-MCD as well as pure IgAN. In a previous study, the median intensity of Gd-IgA1 staining in lupus nephritis specimens was 0+; however, some specimens showed weakly positive Gd-IgA1 staining [19]. In another study comparing IgAN, non-IgA glomerular disease, and healthy controls, serum Gd-IgA1 level was also measured in patients with non-IgA glomerular disease and healthy controls. However, the Gd-IgA1 levels in the patients with non-IgA glomerular disease and healthy controls were significantly lower than those of IgAN patients [20]. Although the reason behind this is not yet completely understood, Gd-IgA1 is produced in conditions other than IgAN, and it can be deposited focally on some glomeruli of these patients. KM55 is an antibody specific for the GalNAc-modified unique epitope on the hinge region of IgA1 [21]. Compared to the lectin-based assays, KM55 immunostaining is a validated and robust method for the detection of serum Gd-IgA1 levels [20]. Nevertheless, since the Gd-IgA1 in lupus patients was negative in most glomeruli, it is more likely to be passive entrapment than an actual inflammatory response. 

In our study, since the follow-up period of the IgAN-MCD cohort was relatively shorter than that of the MCD cohort, the follow-up period was limited to 15 years according to IgAN-MCD. It is possibly because the pathological diagnosis of IgAN-MCD was mainly made after the 2000s. Therefore, some IgAN-MCD patients may have been included among patients diagnosed with MCD before 2000 when the diagnosis of IgAN-MCD was not universal. Despite the abnormal deposition of Gd-IgA in patients with IgAN-MCD, the clinical course of these patients was very similar to that of those with MCD. It is still debatable whether this Gd-IgA deposition plays a pathophysiological role in IgA-MCD. In this study, we excluded patients with endocapillary proliferation, segmental sclerosis, and crescents on biopsy to investigate the clinical manifestations and prognosis of these unusual cases of IgAN-MCD. The present study suggests that Gd-IgA deposition in IgAN-MCD is indolent because IgAN-MCD has very similar clinical features and prognosis to pure MCD. In fact, it is well known that a significant proportion of IgAN patients with Gd-IgA deposition show clinically indolent manifestation. These findings suggest that IgAN-MCD is a dual glomerulopathy in which MCD is superimposed on possibly indolent IgAN.

The present study had several potential limitations. First, the number of cases used in the analysis of the prognosis of IgAN-MCD was significantly lower than that of pure MCD. Thus, IgAN-MCD and MCD patients were matched in 1:3 proportion by propensity score matching, and then the prognoses of the two groups were compared. Second, as a study on human-derived specimens, only four specimens of the 46 IgAN-MCD patients were Gd-IgA1 stained, and we did not measure serum Gd-IgA1 levels. Further large-scale studies will be needed to determine the difference in serum Gd-IgA1 levels between IgAN-MCD and primary IgAN patients. Third, in concordance with MCD, there was no detailed information for the therapeutic drugs in the KoGNET database.

Although it is difficult to draw a definitive conclusion because of the rarity of IgAN-MCD, our findings have important implications. IgAN-MCD is not coincidental deposition of IgA in MCD patients, but a dual glomerulopathy with indolent IgAN with Gd-IgA and superimposed MCD. Further research is needed to determine whether the indolent IgA patients are more vulnerable to MCD.

## Figures and Tables

**Figure 1 jcm-09-02619-f001:**
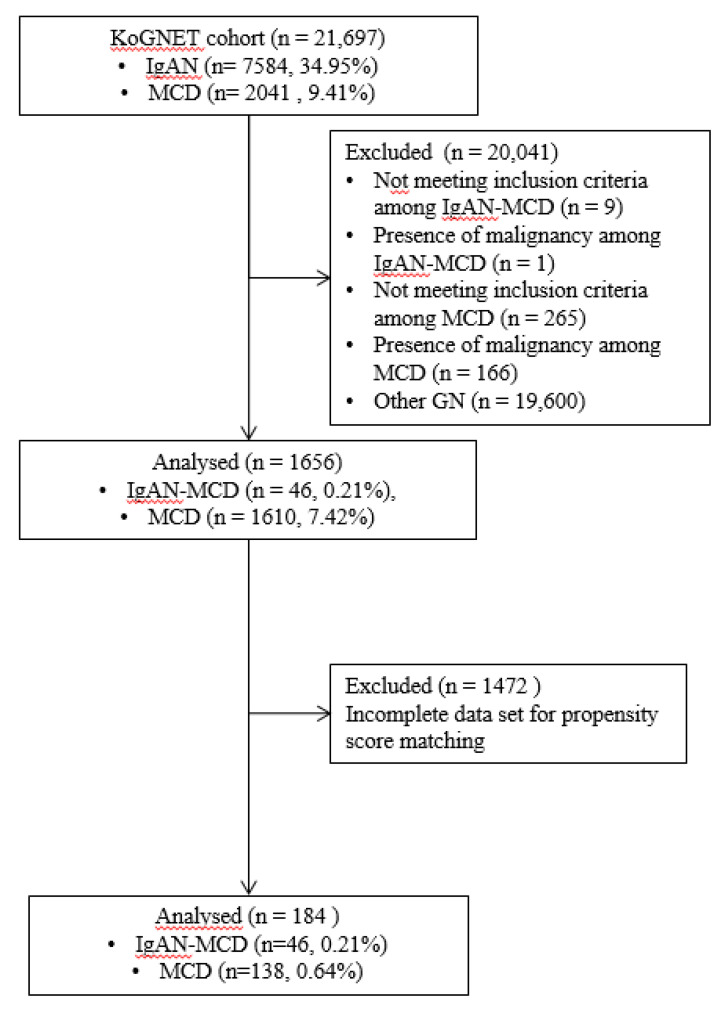
Flow diagram for patient selection and preparation of the propensity score-matched model.

**Figure 2 jcm-09-02619-f002:**
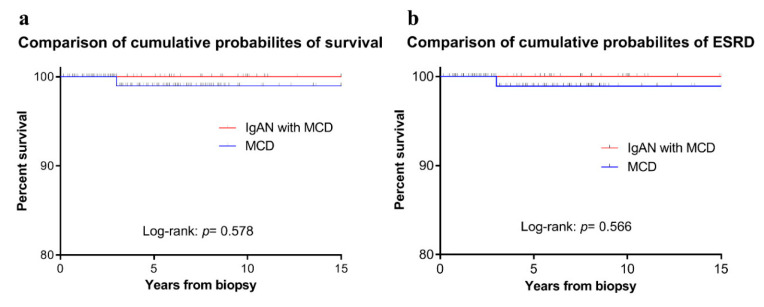
(**a**) Comparison of cumulative probabilities of survival; (**b**) Comparison of cumulative probabilities of end-stage renal disease (ESRD) between IgA nephropathy with minimal change disease (IgAN-MCD) and MCD patients.

**Figure 3 jcm-09-02619-f003:**
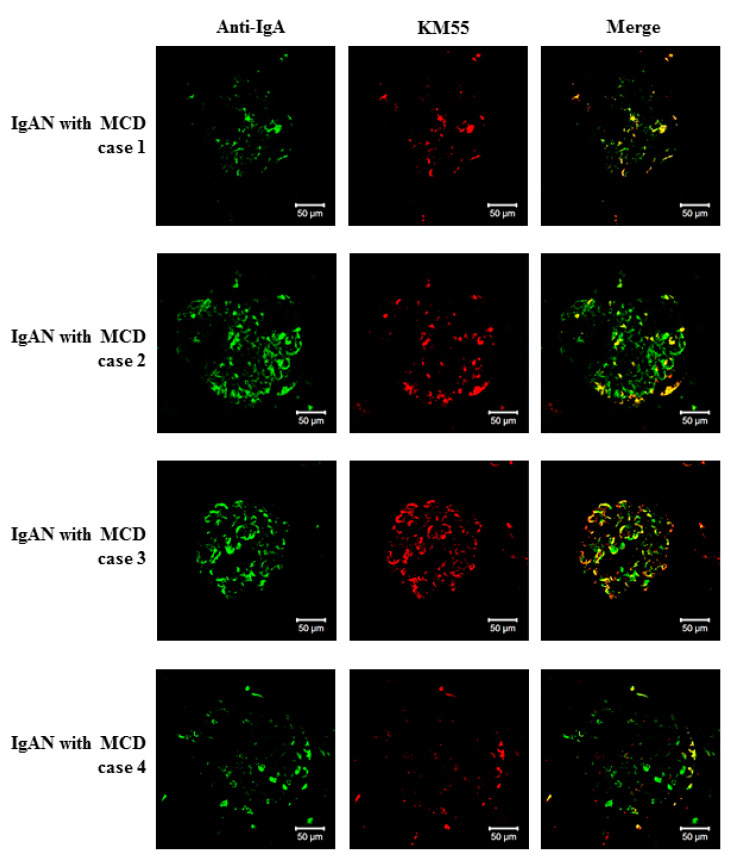
Glomerular deposition of galactose-deficient IgA1 in patients with IgA nephropathy with minimal change disease. Double staining with anti-IgA polyclonal antibody and KM55 monoclonal antibody was performed on kidney specimens from four patients with IgAN-MCD. Four clinical cases are shown. Galactose-deficient IgA1 was localized predominantly in the mesangium with IgA (bars = 50 µm; original magnification ×200).

**Figure 4 jcm-09-02619-f004:**
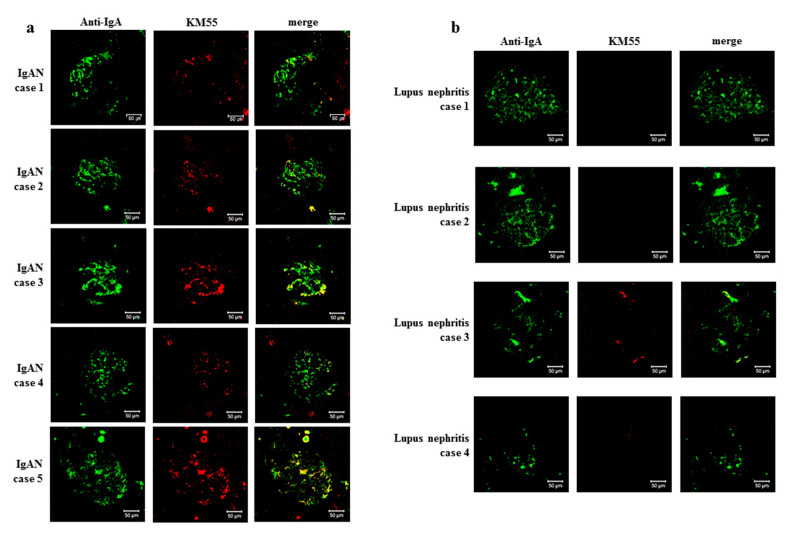
KM55 mAb staining. (**a**) KM55 mAb staining in IgA nephropathy. Glomerular galactose-deficient IgA1 staining was positive in cases of IgA nephropathy (IgAN). (**b**) KM55 mAb staining in lupus nephritis. Four cases of lupus nephritis are shown as examples. In all cases of lupus nephritis that were accompanied by glomerular IgA deposition, glomerular galactose-deficient IgA1 staining was negative (bars = 50 µm; original magnification ×200).

**Table 1 jcm-09-02619-t001:** Characteristics of the study population at baseline.

	Before Matching			After Nearest Neighbor 1:3 Matching		
	IgAN with MCD	MCD	*p* Value	IgAN with MCD	MCD	*p* Value
	(*n* = 46)	(*n* = 1574)		(*n* = 46)	(*n* = 138)	
Age (years)	46.24 ± 18.26	50.02 ± 18.36	0.140	46.24 ± 18.26	46.38 ± 17.94	0.930
Male gender, n (%)	29 (63.0)	974 (61.9)	1.000	29 (63.0)	87 (63.0)	1.000
BMI (kg/m^2^)	25.85 ± 3.59	24.12 ± 4.09	0.007	25.85 ± 3.59	24.79 ± 4.59	0.065
SBP (mmHg)	122.96 ± 17.10	123.72 ± 16.59	0.828	122.96 ± 17.10	125.11 ± 18.68	0.665
DBP (mmHg)	76.62 ± 12.42	77.61 ± 13.14	0.919	76.62 ± 12.42	77.78 ± 13.98	0.906
DM, n (%)	6 (13.0)	82 (5.9)	0.056	6 (13.0)	18 (13.0)	1.000
HTN, n (%)	11 (23.9)	309 (22.1)	0.765	11 (23.9)	36 (26.1)	0.770
CHD, n (%)	1 (2.2)	29 (2.5)	1.000	1 (2.2)	5 (4.6)	0.671
CVD, n (%)	1 (2.2)	19 (1.6)	0.534	1 (2.2)	0 (0.0)	0.294
Serum Albumin (g/dL)	2.22 ± 0.81	2.74 ± 1.16	0.003	2.23 ± 0.81	2.54 ± 1.02	0.051
Serum Cr (mg/dL)	0.86 ± 0.27	0.96 ± 0.36	0.156	0.86 ± 0.27	0.91 ± 0.34	0.544
Total cholesterol (mg/dL)	362.33 ± 131.34	336.76 ± 153.55	0.135	362.33 ± 131.34	342.01 ± 149.83	0.298
LDL cholesterol (mg/dL)	214.68 ± 133.55	217.08 ± 125.82	0.808	214.68 ± 133.55	211.76 ± 114.86	0.920
Serum uric acid (mg/dL)	6.23 ± 1.81	6.15 ± 1.82	0.735	6.21 ± 1.62	6.48 ± 1.64	0.370
UPCR g/g	7.45 ± 5.05	6.04 ± 5.33	0.032	7.45 ± 5.05	7.60 ± 6.00	0.923
UACR g/g	5.09 ± 3.32	4.18 ± 4.17	<0.001	4.63 ± 3.40	4.92 ± 4.66	0.962

Values are presented as mean ± standard deviation or n (%).

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
