# Peer review of "Characterization of IgA Deposition in the Kidney of Patients with IgA Nephropathy and Minimal Change Disease"

_jcm, 2020, doi:10.3390/jcm9082619_

Round 1

Reviewer 1 Report

This retrospective cohort study of IgAN-MCD from almost 22000 patients with GN from Korea compares clinical outcomes between IgAN-MCD and propensity-score matched MCD cohorts, and gd-IgA1 deposition in IgAN-MCD, IgAN and LN. They demonstrate similar clinical characteristics and outcomes between IgAN-MCD and matched MCD patients, but similar gd-IgA1 deposition between IgAN-MCD and IgAN. It is suggested that IgAN-MCD represents both MCD and indolent IgAN, as opposed to coincidental IgA deposition in MCD.

Overall, I think this is a well conducted study that provides insight as to outcomes and pathogenesis of IgAN-MCD and I support its publication. It's a shame information on medications used in the MCD and IgAN-MCD groups is unavailable. I suggest two additions to improve the article.

  1. Add explanation as to why the MCD cohort follow-up is much longer than IgAN-MCD (shown in Figure 2)
  2. Are LM and EM images available from the IgAN-MCD cases tested with IF in Figure 3? If so, can they be included to demonstrated whether IgAN-MCD is similar in histologic morphology to MCD? 

Author Response

We appreciate the efforts and time taken by the reviewers in reviewing our manuscript. Please see the attachment

Reviewer 2 Report

This is an interesting paper that studies the connection between IgA-Nephropathy and Minimal Change Disease, defining a dual glomerulopathy where MCD is superimposed on indolent IgAN. Previous studies have reported these patients to be suffering from a variant type of IgAN, while others have reported this condition as a coincidental deposition of IgA in patients of MCD.

The design of this retrospective study is well developed and the topics of the work add a valuable contribution to the field. This study can stimuli further research to establish the implication of this dual glomerulopathy. the only issue is the small number of patients with this dual glomerulopathy. 

Minor modifications: 

Line 34 change "characterization" in "characterizations"
Line 51 change "coincidental" in "a coincidental".

Author Response

(The authors gave the same response as above.)

Reviewer 3 Report

The authors have done a retrospective study on a area of glomerular disease where we have very little knowledge.

They have tried to clarify if IgA-MCD is variant of MCD or is it a dual pathology. They have excluded patients with aggressive IgA features such as crescents, endocapillary proliferation, sclerosis etc.

They have shown the prognosis of IgA MCD is comparable to MCD, similar to previous publication.

The biggest drawback I see is that they only chose 4 patients with IgA-MCD for staining of Gd-IgA1, however they have highlighted that in their results and recommended further larger studies.

Author Response

(The authors gave the same response as above.)
